# Cyclic Glycine-Proline (cGP) Normalises Insulin-Like Growth Factor-1 (IGF-1) Function: Clinical Significance in the Ageing Brain and in Age-Related Neurological Conditions

**DOI:** 10.3390/molecules28031021

**Published:** 2023-01-19

**Authors:** Jian Guan, Fengxia Li, Dali Kang, Tim Anderson, Toni Pitcher, John Dalrymple-Alford, Paul Shorten, Gagandeep Singh-Mallah

**Affiliations:** 1Department of Pharmacology and Clinical Pharmacology, Faculty of Medicine and Health Sciences, School of Biomedical Sciences, The University of Auckland, Auckland 1142, New Zealand; 2Centre for Brain Research, Faculty of Medicine and Health Sciences, School of Biomedical Sciences, The University of Auckland, Auckland 1142, New Zealand; 3Brain Research New Zealand, The Centre for Research Excellent, Dunedin 9016, New Zealand; 4The cGP Lab Limited New Zealand, Auckland 1021, New Zealand; 5Guangdong Pharmaceutical University, Guangzhou Higher Education Mega Center, Guangzhou 510075, China; 6The Seventh Affiliated Hospital, Sun Yat-sen University, Shenzhen 518107, China; 7Shenyang Medical College, Shenyang 110034, China; 8New Zealand Brain Research Institute, Christchurch 4710, New Zealand; 9Department of Medicine, University of Otago, Dunedin 9016, New Zealand; 10Department of Neurology, Canterbury District Health Board, Christchurch 4710, New Zealand; 11Department of Psychology, University of Canterbury, Christchurch 4710, New Zealand; 12AgResearch Ltd., Ruakura Research Centre, Hamilton 3214, New Zealand; 13Riddet Institute, Massey University, Palmerston North 4474, New Zealand; 14Institute of Neuroscience and Physiology, Sahlgrenska Academy, University of Gothenburg, 40530 Gothenburg, Sweden

**Keywords:** insulin-like growth factor-1 (IGF-1) function, cyclic glycine-proline (cGP), ageing brain, stroke, Alzheimer’s disease, Parkinson’s disease

## Abstract

Insulin-like growth factor-1 (IGF-1) function declines with age and is associated with brain ageing and the progression of age-related neurological conditions. The reversible binding of IGF-1 to IGF binding protein (IGFBP)-3 regulates the amount of bioavailable, functional IGF-1 in circulation. Cyclic glycine-proline (cGP), a metabolite from the binding site of IGF-1, retains its affinity for IGFBP-3 and competes against IGF-1 for IGFBP-3 binding. Thus, cGP and IGFBP-3 collectively regulate the bioavailability of IGF-1. The molar ratio of cGP/IGF-1 represents the amount of bioavailable and functional IGF-1 in circulation. The cGP/IGF-1 molar ratio is low in patients with age-related conditions, including hypertension, stroke, and neurological disorders with cognitive impairment. Stroke patients with a higher cGP/IGF-1 molar ratio have more favourable clinical outcomes. The elderly with more cGP have better memory retention. An increase in the cGP/IGF-1 molar ratio with age is associated with normal cognition, whereas a decrease in this ratio with age is associated with dementia in Parkinson disease. In addition, cGP administration reduces systolic blood pressure, improves memory, and aids in stroke recovery. These clinical and experimental observations demonstrate the role of cGP in regulating IGF-1 function and its potential clinical applications in age-related brain diseases as a plasma biomarker for—and an intervention to improve—IGF-1 function.

## 1. Introduction

### 1.1. Decline in IGF-1 Function with Age and Its Association with Age-Related Conditions

As an endogenous hormone, IGF-1 is essential for brain development, cognitive function, energy metabolism, and wound healing [1,2,3,4,5,6]. In addition to its function as a neurotrophic factor in brain, IGF-1 regulates cerebral and systemic vascular remodelling and function [7,8]. The growth hormone (GH) controls the liver production of IGF-1 [9], which declines with ageing to a very low measurable level in people over 60 years of age [9,10]. An age-related deficiency in IGF-1 production and function can contribute to cardiovascular aging [11], as well as cerebrovascular diseases such as stroke [5,12,13], hypertension [6,14], and cognitive impairment [15,16,17].

### 1.2. Regulation of Bioavailablility of IGF-1 in Circulation

The majority of IGF-1 is present in circulation in either a binary complex of IGF-1/IGFBP-3 or a ternary complex of IGF-1/IGFBP-3/acid-labile subunit (ALS) [18]. The ternary complex can increase the half-life of circulating IGF-1 by up to 12 h [18]. These binary and ternary complexes act as reservoirs to protect circulating IGF-1 from being metabolized [19]. IGF-1 function is mediated through the activation of IGF-1 receptors, which, in turn, initiate the downstream signalling pathways, such as the PI3k-AKT signal transduction pathway [20,21]. However, the binding of IGF-1 to IGFBP-3 prevents IGF-1 from activating the IGF-1 receptors due to a higher binding affinity to the IGFBP-3 than to the IGF-1 receptors [22]. Thus, the majority of IGF-1 in circulation is not bioavailable, but the binding of IGF-1 to IGFBP-3 is reversible, which serves to regulate the amount of bioavailable IGF-1 and thereby, the homeostasis of IGF-1 function in circulation [23,24,25]. The ALS only interacts with the binary complex and does not bind to free IGF-1 or free IGFBP-3. The ternary complex does not prevent the catabolism of IGFBP-3 or the release of IGF-1 from the binary complex [18]. Thus, ALS binding may not influence the reversible binding of IGF-1 to IGFBP-3.

The reversible binding of IGF-1 to IGFBP-3 regulates the amount of bioavailable IGF-1 in circulation. This is more critical following the decline in IGF-1 production with age and in neurological conditions associated with IGF-1 deficiency. Characterised as IGF-1 resistance, the increase in IGF-1 concentration has been shown in age-related degenerative conditions, such as Alzheimer’s disease and Parkinson’s disease [26,27,28]. This associates with the increase in IGFBP concentrations, which reduces the amount of bioavailable IGF-1 in circulation and in brain tissue. Recent research has also demonstrated the involvement of cGP in the regulating of the bioavailability of IGF-1 [29], as discussed below.

## 2. cGP Is a Bioactive Compound

The following section addresses the discovery and research development of cGP as a endogenous neuropeptide.

### 2.1. Formation of Endogenous cGP

Free IGF-1 is not enzymatically stable [25]. The enzymatic breakdown of IGF-1 at its N-terminal forms des-N-(1-3) IGF-1 (des-IGF-1) in plasma and brain tissue [30,31]. As a major binding site for IGFBPs, the loss of the N-terminal tripeptide, Glycine-Proline-Glutamate (GPE) reduces the binding affinity of des-IGF-1 for IGFBPs [22]. IGFBPs treatment reduces the formation of GPE by increasing IGF-1 binding [32]. The different rotation in the prolyl–glycine bond determines the trans or cis isoform GPE, with a constant 4:1 ratio of trans/cis isoforms in circulation [33]. Both isoforms of GPE are also enzymatically unstable [34,35]. The cis isoform of GPE forms cGP through cyclisation after the enzymatic cleavage of glutamate [33]. By evaluating the amount of GPE in plasma and brain tissues in rats, Baker and others described the enzymatic degradation of GPE [35]. Unlike the linear isoforms of GPE, such a cyclic structure may render cGP resistant to enzymatic breakdown and become more lipophilic for better tissue uptake [33,36].

### 2.2. cGP as a Bioactive Peptide

Following the isolation of cGP from rat brain tissue [37], the neurological function of cGP and its analogues have been extensively examined by different researchers. Gudasheva et al. reported the efficacy of cGP and its analogues in protecting the brain from ischemic brain injuries and improving cognitive function [38,39,40]. Experimental research shows that both GPE and cGP exhibit pharmacological effects similar to those of IGF-1. For example, the administration of GPE, or cGP at a dose equimolar to those of IGF-1 protects the brain from hypoxic-ischemic injury in the same experimental setting in rats [34,41,42,43]. As a transient intermediate between IGF-1 and cGP, the biological effects of GPE may be mediated through cGP. cGP is a small, lipophilic, enzymatic stable peptide. It is orally bioavailable with effective tissue (brain) uptake [43,44,45,46].

As part of its pharmaceutical development, a structural analogue of cGP, cyclic Gly-2allyl-Pro, protects the brain from ischemic injury, 6-OHDA-induced motor deficit, and scopolamine-induced acute memory impairment [42,47,48]. Clinical trials of cGP analogues showed promising outcomes for treating developmental neurological conditions [49,50,51,52], including Rett syndrome and Fragile X syndrome [49,53,54]. The neuroprotective effects of cGP have been shown to be associated with promoting neurogenesis, synaptic function, and vascular remodelling, and inhibiting inflammation, apoptosis, and vascular damage. However, these changes could be a result, rather than the cause, of reduced brain damage. Several receptors have been suggested to mediate the biological function of GPE and cGP; for example, α-amino-3-hydroxy-5-methyl-4-isoxazolepropionic acid receptor, γ-aminobutyric acid (, *N*-methyl-D-aspartate receptor, and metabotropic glutamate receptors [38,55,56,57], but the results are inconclusive. It is also known that the N-terminal of IGF-1 does not interact with the IGF-1 receptors [57,58]. The investigations into the mechanism of cGP led to the discovery of its role in regulating IGF-1 function and its association with age-related neurological conditions.

## 3. Mode of Action of cGP

Recent research results have provided the initial evidence for the mode and mechanism of cGP action, in which cGP action is mediated by normalising IGF-1 bioavailability, and thus its function. The effects of cGP in IGF-1 function are first evaluated by examining the survival/growth and tube formation of human endothelial cells [29]. The results from several experimental paradigms reveal that treatment with cGP can either promote, maintain, or inhibit IGF-1-induced cell survival/growth when IGF-1 treatment alone fails to stimulate, or moderately or highly stimulates, cell survival/growth [29]. The presence of IGF-1 is essential for cGP to be effective in experimental settings. The overexpression or knockdown plasmids of IGF-1 receptors has further confirmed that the efficacy of cGP in endothelial cells is mediated via IGF-1 and is a result of the regulated IGF-1 effect [29]. To maintain the homeostasis of IGF-1 function, cGP stimulates IGF-1 function when IGF-1 function is insufficient, or inhibits IGF-1 function when IGF-1 is overly promoted, without altering the function of IGF-1 within a physiological range. The different effects of cGP on endothelial cell survival/growth are dependent on its concentration relative to that of IGF-1, in which a higher cGP/IGF-1 molar ratio leads to a stimulatory effect, whereas a lower cGP/IGF-1 ratio results in an inhibitory effect. cGP does not alter IGF-1-induced cell growth/survival when the concentration of cGP is similar to that of IGF-1 [29]. Experimental studies have also demonstrated vascular effects of IGF-1 [7,59], cGP [29], or GPE, a transient intermediate between IGF-1 and cGP [60,61], in preventing vascular damage and improving vascular remodelling.

As a metabolite from a major binding site of IGF-1, cGP retains the binding affinity to IGFBP-3, thus competing with IGF-1 for the binding of IGFBP-3. Concentration dependent competitive binding between cGP and IGF-1 to IGFBP-3 is evaluated using an in vitro peptide–peptide interaction assay [29]. When IGFBP-3 is incubated with different concentrations of cGP and IGF-1 in differing ratios, a higher ratio of cGP/IGF-1 increases the percentage of unbound/total IGF-1, whereas a lower ratio of cGP/IGF-1 reduces the percentage of unbound/total IGF-1 [29,62]. These data suggest that the cGP can interfere with IGF-1 binding to IGFBP-3, and the ratio of cGP/IGF-1 represents the amount of free IGF-1 [29]. This hypothesis has been subsequently confirmed by experimental studies and clinical observations.

## 4. Clinical Relevance

In general, the clinical relevance of cGP is in those conditions associated with IGF-1 deficiency, particularly the age related vascular diseases—metabolic hypertension, stroke, age-related cognitive decline, and some neurodegenerative disorders such as Parkinson’s disease (PD) [63,64] and Alzheimer’s disease (AD) [65,66,67]. Given the role of cGP in regulating IGF-1 function, we evaluated the changes in cGP, IGF-1, and IGFBP-3 concentrations in plasma and brain tissues, as well as their association with clinical outcomes in patients with metabolic hypertension, stroke, age-related cognitive impairment, PD, and AD. To support the interpretation of these clinical observations, the efficacy of cGP has been evaluated in animal models of metabolic disorders, stroke, memory impairment, and PD.

### 4.1. Hypertension

Obesity, dyslipidemia, hyperglycemia, and hypertension are collectively defined as metabolic syndrome, with a high risk of developing cardiovascular and cerebrovascular diseases [68] and dementia. IGF-1 plays an important role in energy metabolism [69,70]. IGF-1 deficiency is associated with obesity [71] and hypertension [72]. This age-related condition can also occur in a younger population, i.e., gestational hypertension. In healthy women, plasma concentration IGF-1 decreases transiently during early pregnancy due to GH resistance, then recovers after the women give birth, but it remains low in women with metabolic syndrome [73]. This difference provides a window of opportunity to evaluate whether cGP changes by responding to IGF-1 concentration in humans. A cross-sectional study compared the changes in plasma concentrations of cGP, IGF-1, and IGFBP-3 between healthy women and women with hypertension 6 years post-partum [73]. Plasma concentrations of IGFBP-3 and cGP are lower in hypertensive women compared to those with normal blood pressure, independent of obesity status [73]. The reduced plasma IGFBP-3 concentration is an endogenous response to improve the bioavailability of IGF-1, whereas the lowered plasma cGP can reduce the bioavailability of IGF-1 in circulation. To support this interpretation, we examined the efficacy of cGP in a rat model of metabolic syndrome. The rats develop metabolic syndrome after 8 weeks of high-fat diet feeding (HFD) during post-natal weeks 3–11 [74]. Compared to the group on the control diet, the HFD group shows an increase in systolic blood pressure (SBP), adiposity, and insulin insensitivity. As noted in the observations in women, HFD feeding decreases the concentrations of IGF-1 and IGFBP-3, whereas the cGP concentration increases in the plasma [74], this being a compensatory response to improve IGF-1 function. The administration of cGP from post-natal weeks 11–15 reduces the SBP and retroperitoneal fat weight, but not body weight and insulin resistance [74]. Correlation analysis shows that the rats with higher plasma cGP concentrations have a lower retroperitoneal fat weight, independent of the status of obesity. Endogenous concentration of cGP is positively correlated with SBP in saline-treated hypertensive rats [74], which could be an endogenous, but ineffective, response to improve IGF-1 function. With increasing plasma cGP concentration, via administration, hypertensive rats with higher cGP showed lower SBP [74]. This effect was not seen in normotensive rats [74]. Treatment with cGP that increases plasma cGP concentration leads to an effective response in normalising blood pressure and retroperitoneal fat weight. These beneficial effects of cGP on SBP and retroperitoneal fat suggest a therapeutic potential for cGP in cardio-metabolic complications including, for example, stroke.

Hypertension is a life-long risk factor for stroke [75] and shares the pathology of endothelial dysfunction and the pathophysiology of impaired IGF-1 function [36]. The following section reviews the role of cGP in regulating IGF-1 function at the onset of stroke and during first 3 months of recovery as a neurological condition of vascular origin [13,76].

### 4.2. Stroke

The function of IGF-1 in stroke recovery has been well-documented [5]. Most stroke patients make a partial recovery in function within the first 3 months, a critical period for ‘self-made’ (i.e., spontaneous) recovery [77]. To investigate whether endogenous cGP is associated with this self-made recovery in stroke patients, we evaluated plasma concentrations of cGP, IGF-1, and IGFBP-3 at the onset of stroke and 3 months post-stroke recovery [36]. This longitudinal clinical study includes 50 non-stroke control participants and 34 stroke patients. The National Institutes of Health Stroke Scale (NIHSS) is assessed within three days of hospital admission (<3 days) as a baseline, and at days 7 and 90; the modified Rankin Scale (mRS) and Fugl-Meyer Upper-Limb Assessment Scale (FM-UL) are administered at days 7 and 90. Plasma samples were are also collected from patients at baseline, and days 7 and 90.

Compared with the control participants, the plasma concentrations of IGFBP-3 and cGP were lower in stroke patients at the time of hospital admission, and the concentration of IGF-1 was similar [36]. The reduction of plasma IGFBP-3 in stroke patients could be a positive response to increase the amount of bioavailable IGF-1 in the circulation, whereas the low plasma cGP suggests a reduction of bioavailable IGF-1 in circulation during the onset of stroke [36]. Multiple regression analysis after adjustment for potential confounding factors (age and baseline NIHSS scores [78]) suggests that the patients with a higher plasma concentration of cGP and/or molar ratio of cGP/IGF-1 at the time of hospital admission make a better recovery, with fewer neurological deficits at day 90 post-stroke.

Plasma concentrations of cGP and cGP/IGF-1 molar ratio increase over 90 days, in parallel with the improved motor function and clinical outcomes. However, the concentrations of IGF-1 and IGFBP-3 remain the same during the recovery phase [36]. Several groups have evaluated the changes in the plasma concentration of IGF-1 at 3 days after stroke, with inconsistent results [79,80]. Changes in plasma IGF-1 concentrations at the one-set of stroke appear to be associated with mortality, but not functional recovery [81]. This observation also supports the notion that changes in plasma IGF-1 concentrations are not associated with IGF-1 function during stroke recovery, whereas cGP-related changes (cGP/IGF-1 molar ratio) are [36]. This association between cGP/IGF-1 ratio and stroke recovery provides additional evidence for cGP as a regulator of IGF-1 function.

This idea is also supported by experimental studies. Intracerebroventricular administration of cGP (0.2 µg/rat, *i.c.v.*) 2 h after a unilateral hypoxic-ischemic brain injury partially reduces neuronal loss in the hippocampus, as compared to that noted in the control rats. In an analysis of capillary densities, we found almost complete vascular protection/restoration in the hippocampus after cGP treatment [29]. This treatment effect on vascular protection is in parallel to an increase in the phosphorylation of IGF-1 receptors in the capillaries [29]. The expression of IGF-1 receptors does not occur in the endothelial cells, but possibly on the pericytes [29], which play a key role in vascular remodelling through expressing angiogenic factors [82]. Taken together, the data suggest that the cGP-associated vascular protection is mediated through IGF-1. The treatment with cGP did not further increase vascular density in the control hippocampus, where there is no vascular damage [29]. The administration of a structural analogue of cGP six hours after the onset of stroke protects the brain from ischemic injury and improves long-term sensory-motor function [42].

Changes in the cGP/IGF-1 molar ratio may assist in the prediction of the clinical outcome and management of stroke [36]. The molar ratio of cGP/IGF-1 in plasma during the onset of stroke may be developed as a biomarker for predicting the ability of functional recovery in stroke patients. Considering the oral bioavailability and dynamic central uptake of cGP in humans [45], the development of this biomarker could provide guidance to assist future clinical trials in stroke.

Age-related cognitive impairment has been suggested as being associated with vascular degeneration and poor vascular function [17]. In the next section, we review the changes in the cGP/IGF-1 molar ratio in age-related cognitive status.

### 4.3. Age-Related Cognitive Decline

Healthy vascular function is critical for maintaining normal cognition in humans [16,83]. Age-related cognitive decline is considered to be associated with the vascular ageing processes [17]. Aged rats show poor spatial memory compared to young rats in Morris Water Maze performance [17]. Stereological analysis shows that the neuronal density in the hippocampus is similar in young and aged rats [17]. Instead, the aged rats that show memory impairment have significantly fewer capillaries and glial cells, as well as a lower expression of synaptic markers in the hippocampus, than do the young rats [17]. These data suggest that age-related cognitive impairment is associated with neuronal dysfunction that is due, at least in part, to glial and vascular degeneration.

Vascular ageing contributes to cognitive decline in humans [84]. IGF-1 plays an essential role in the vascular remodelling of the adult brain [7] and supports the retention of normal cognition [85]. Age-related IGF-1 deficiency contributes to cognitive impairment in older people [85,86,87,88]. A recent observational study examined the association between plasma concentrations of IGF-1, IGFBP-3 and cGP and the cognitive scores in a group of older people (mean = 74.5 years of age) with normal cognitive function. Multiple regression analysis after adjustment for age shows that the cGP concentration and the cGP/IGF-1 molar ratio are positively associated with scores on the Montreal Cognitive Assessment (MoCA), a global measure of cognitive function derived from comprehensive neuropsychological testing (Global Z), and test scores in the learning and memory domain (LMD) [89]. In contrast, plasma concentration of IGF-1 and IGFBP-3 are not associated with these cognitive measures [89]. These data suggested that older people with higher plasma cGP concentrations and/or a higher cGP/IGF-1 molar ratio exhibit better learning and memory functions. An independent study observed an increase in the cGP/IGF-1 molar ratio and a decrease in the IGFBP-3 concentration in older people with mild cognitive impairment (Figure 1) [90], suggesting an ineffective response to overcome age-related cognitive decline.

### 4.4. Parkinson’s Disease (PD)

Using immunohistochemical staining to visualise endothelial cells, Guan et al. first reported endothelial degeneration in human PD brains [91]. The loss of endothelial cells in the capillaries leads to the formation of string vessels with no function in cerebral circulation [92].

Vascular degeneration in older people and PD is associated with IGF-1 deficiency, which also impairs the vascularisation processes [27]. Thus, vascular degeneration may be, at least partially, an age-related pathology which can potentially increase the risk of developing cognitive impairment in older people [93,94,95] and PD patients [3,86,96,97].

While plasma IGF-1 decreases with age [86], it is elevated in PD patients compared to non-PD controls [98,99,100]. This age effect confounds the PD effects on the IGF-1 concentration in circulation. To reveal the specific association of IGF-1, cGP, and IGFBP-3 with age, cognitive function, or motor deficits, the data are analysed using a multiple linear regression model, with adjustment for age, motor, and cognitive scores accordingly. The analysis suggested that the changes in the plasma concentration of IGF-1 and cGP with age are associated with cognitive status [89], and the changes in plasma IGFBP-3 concentrations are associated with motor deficits in PD.

The plasma concentration of IGF-1 decreases with age, whereas cGP increases with age in the PD patients with normal cognition (PD-N), leading to an age-related increase in the cGP/IGF-1 molar ratio [89]. This increase in the cGP/IGF-1 molar ratio is absent in the PD group with mild cognitive impairment (PD-MCI). In contrast, the cGP/IGF-1 molar ratio decreases with age in the PD group with dementia (PD-D). Compared with the PD-N group, the association regression slope of the cGP/IGF-1 molar ratio is reversed in the PD-D group. These data suggest that cognitive impairment in PD is age related, and the age effects on the cGP/IGF-1 molar ratio are differently associated with the cognitive status in PD [89].

An increase in the plasma cGP/IGF-1 molar ratio with age may contribute to the preserved cognitive function in the PD-N group, possibly due to the improvement in the amount of bioavailable IGF-1 in the plasma. In contrast, the decrease in the cGP/IGF-1 molar ratio with age in the PD-D group may reverse this effect during the progression to dementia. The static relationship between the cGP/IGF-1 molar ratio and age in the PD-MCI group could be a transition phase prior to dementia. These observations raise the possibility that the association between the cGP/IGF-1 molar ratio and age may assist in predicting cognitive status and the risk of advanced cognitive impairment in PD patients, but these hypotheses require further confirmation with longitudinal observations.

IGF-1 resistance, characterised as an increase in IGF-1 concentration, may contribute to the progression of motor deficits in PD. The same study that evaluated the cognitive status also examined the association of cGP, IGF-1, and IGFBP-3 with UPDRS, commonly used for evaluating motor deficits in PD patients. The IGF-1 concentration in plasma is higher in PD patients than in the non-PD controls, suggesting IGF-1 resistance in this cohort of patients. Multiple linear regression analysis after correction for age and 3 individual cognitive scores shows a positive correlation between UPDRS and the plasma concentration of IGFBP-3 (Table 1, unpublished data). The motor deficits in PD are not correlated with the cGP and IGF-1 concentrations.

High concentration of IGFBP-3 would increase IGFBP-3 binding to IGF-1 and reduce the amount of bioavailable IGF-1 in circulation. Given that only bioavailable IGF-1 can cross the blood–brain barrier, the reduction in bioavailable IGF-1 in plasma can also reduce the brain penetration of IGF-1, which is the main source of IGF-1 in brain tissues. Provided that cGP can displace IGF-1 from IGFBP-3 binding, increasing cGP through an intervention might also reduce IGF-1 resistance and increase the bioavailability of IGF-1 in circulation. Using immunohistochemistry, Yang et al. examined the expression of IGF-1 and IGFBP-2 in the brain regions of PD cases. Compared to non-PD cases, the expression of IGF-1 is reduced, whereas the expression of IGFBP-2, a main IGFBP, is increased in PD brain tissues [27]. The higher expression of IGFBP-2 is collocated with glial cells, suggesting an association with brain inflammation in PD [27].

### 4.5. Alzheimer’s Disease (AD)

Vascular aging has been suggested to contribute to the progression, or even the cause, of AD [101], a neurodegenerative condition marked by cognitive impairment and dementia. Wang (2019) compares the changes in IGF-1, cGP, and IGFBP-3 between 15 patients with mild AD and 15 normal controls. The mean Addenbrookes Cognitive Examination (ACE-111) score is 94.21 ± 0.72 in cognitively healthy participants and 82.67 ± 1.45 in those who had a diagnosis of AD (*p* < 0.001). The mean value for the ACE-111 score in the AD patients suggests that their dementia status is relatively mild. Compared to non-AD controls, the AD patients exhibit an increase in cGP and a decrease in IGFBP-3 concentrations, while changes in IGF-1 are not significant [90]. These results also suggest a collective regulatory response between cGP and IGFBP-3 to maintain IGF-1 function and to slow down disease progression to dementia.

While the majority of bioavailable IGF-1 in brain tissues is transported from the circulation to brain tissues by the activation of IGF-1 receptors [102], a small amount is produced locally through glial cells, which themselves can be enhanced after brain injury [12,27]. Even though IGF-1 in brain tissues is more bioavailable than that in circulation, the reversible binding of IGF-1 to mainly IGFBP-2 also plays a role in influencing the bioavailability of IGF-1 in brain tissues. While cGP can be generated locally via IGF-1 metabolism in brain tissues, the majority of cGP is transferred across the blood–brain barrier (BBB) from circulation [45,74].

A recent human brain tissue study evaluated changes in IGF-1, cGP, and IGFBP-2 and -3 in the brain regions of AD cases and age-matched control cases [26]. The concentration of total IGF-1 is lower in the inferior-frontal gyrus and middle-frontal gyrus of the AD brains compared to the control brains. Vascular degeneration is a key pathological feature of AD [103]. As part of the structure and function of the BBB, the degeneration of capillaries in the AD brain [104] may reduce IGF-1 receptors, which are essential for central transfer of bioavailable IGF-1. Given the essential role of circulating IGF-1 in vascular remodeling [7,85], a deficiency of plasma IGF-1 can impact vascular remodelling and the function of the BBB in transferring IGF-1 from plasma to brain tissues.

In contrast, the concentrations of IGFBP-3 and cGP are higher in the inferior-frontal gyrus and middle-frontal gyrus in the AD compared to the control cases [26]. The increase in IGFBP-3 concentration, which could be a result of an inflammatory response to brain degeneration [105], may reduce bioavailable IGF-1 in the AD brain, further reducing the amount of bioavailable IGF-1. The increase in cGP is likely a response to improve the bioavailability of IGF-1 in the AD brain [26].

Given the anabolic effects of IGF-1 in the brain [106], the protein concentration could be an indication of IGF-1 function in brain. The authors analyzed the total protein concentration of the brain tissues used for analyzing cGP, IGF-1, and IGFBPs. The total protein concentration in AD brain tissues is lower [26] than in the controls, and is negatively associated with IGFBP-3 in AD cases. In contrast, the total protein concentration is associated with a decrease in IGF-1 concentration, but an increase in cGP concentration in the control cases. These observations suggest that IGF-1 function is deficient in the AD brain. The increase in IGFBP-3 in the AD brain could be the result of inflammation, and the increase in cGP could be a positive response to age-related decline in IGF-1. This interpretation is supported by other experimental observations [44,46]. High-fat diet feeding after weaning induces metabolic syndrome in rats, with reduced synaptic expression in the hippocampus and striatum [46]. The peripheral administration of cGP increased the cGP concentration in brain tissues and normalized synaptic expression, without increasing IGF-1 concentration in the brain. Thus, the effects of cGP on synaptic function could be associated with an increase in bioavailable IGF-1 in the brain tissue [46]. Central administration of cGP reduces brain damage in rats with hypoxic-ischemic injury by activating the IGF-1 receptors [29]. Mediated through developmental programming, the memory improvement of adult offspring after maternal administration of cGP to lactating rat dams is associated with advancing astrocytic plasticity and vascular remodelling, leading to synaptic trafficking through the glutamine–glutamate cycle in the hippocampus (Figure 2) [44,107]. Thus, the memory improvement associated with cGP intervention could be, in part, a vascular effect.

### 4.6. A Plasma Biomarker for Age-Related Neurological Diseases

Vascular disease is a common risk factor contributing to cognitive impairment and stroke [15,108,109]. The normal function of IGF-1 is essential for cerebral vascular remodelling in mature brain and is mediated through the endothelium of cerebral vessels [7]. Given the association of vascular disease with the function of circulating IGF-1, a plasma biomarker for IGF-1 function would have clinical significance in assisting the assessment of the status and prognosis of vascular disease. Plasma concentrations of IGF-1 and the ratio of IGF-1/IGFBP-3 have been under clinical evaluation for their association with vascular diseases, i.e., age related cognitive status and stroke recovery [1,96,110]. The findings in this area have been mixed and difficult to interpret [1,96,110]. As discussed earlier, the majority of IGF-1 in plasma is not bioavailable, so it does not represent the functional impact of circulating IGF-1. The IGF-1/IGFBP-3 ratio could be a better representation of functional IGF-1 in circulation than IGF-1 alone, but more than 70% of circulating IGFBP-3 is independent of IGF-1. Given its association with age-related cognitive status and stroke recovery, changes in cGP/IGF-1 molar ratio and/or cGP concentration could be a better plasma biomarker to identify those at greater risk of cognitive decline and poor ability to recover from stroke.

The administration of cGP protects the rat brain from ischemic injury by promoting IGF-1-mediated vascular protection/remodelling [29]. Similarly, cGP treatment normalises systolic blood pressure in rats with metabolic disorders [74]. Given the existence of cerebral vascular degeneration/dysfunction in age-related cognitive impairment [17], the changes in plasma the cGP/IGF-1 molar ratio may be a biomarker for the identification of the window of opportunity for suitable intervention, monitoring treatment response, and individualizing medical treatment.

## 5. Discovery of Natural cGP

New Zealand blackcurrants contain natural cGP [45]. In an open-label trial conducted in 13 PD patients with normal cognitive function, supplementation with the capsulised blackcurrant extract (BCA) resulted in a significant increase in cGP concentration in the cerebrospinal fluid (CSF) of the participants, without altering CSF concentration of IGF-1 and IGFBPs. The concentration of cGP in the CSF is correlated with plasma cGP concentration. These data suggest that BCA capsules provide natural cGP, which is orally bioavailable to the human brain. While there is no improvement in motor function, the Hospital Anxiety and Depression Scale (HADS) and subscale anxiety scores of the participants are reduced after 4 weeks of supplementation [111] (Table 2). It is surmised that these effects are mediated by cGP, since the supplementation increases the concentration of cGP in the plasma and CSF of the PD patients [45].

Patients also scored significantly lower in the emotional well-being domain of the PDQ-39 quality of life measure after BCA supplementation (t_9_ = −3.97, *p* = 0.008). Several clinical trials that have evaluated the efficacy of cGP-related compounds also report the benefits in mood [49,54].

Direct delivery of cGP to CSF activates IGF-1 receptors in the capillaries, leading to improved vascularisation in the brain [29]. Similarly, peripheral administration of cGP improves memory by promoting astrocytic plasticity and vascularisation [107]. Mediated by normalising IGF-1 function, this mechanism of cGP makes the clinical application of cGP particularly attractive. In contrast, the clinical application of IGF-1 is limited due to its short half-life, poor bioavailability in circulation, limited tissue uptake, and more critically, its narrow bell-shaped dose-dependent efficacy with potential for causing adverse effects [41]. Neuren Pharmaceuticals has been developing cGP analogues for treating developmental neurological conditions for more than 20 years (https://www.neurenpharma.com, accessed in November 2022). Information from clinical development has demonstrated the safety for clinical use. If further developed, cGP administration through natural supplements may open an additional and affordable intervention for vascular diseases by improving IGF-1 function in circulation.

## 6. Limitations of the Review and Future Studies

Despite the number of experimental studies and clinical observations discussed herein, research on cGP regulating IGF-1 function is still in an early stage. Since the discovery of the N-terminal metabolism of IGF-1 more than 30 years ago [58], research progress on IGF-1 metabolites has been slow and largely ignored. Data presented in this review provide initial, yet clear, evidence supporting the role of cGP in regulating IGF-1 function and also highlights many research gaps that need to be filled. These gaps include the molecular mechanism that activates the enzymic reaction to form cGP and to break down IGFBPs, the potential role of cGP in endocrine regulation of IGF-1, and the mechanisms beyond competitive binding to IGFBP-3. The clinical relevance of cGP in other vascular diseases and the health benefits of supplementation of cGP through natural foods require further clinical evaluation. We hope this review will stimulate research interest in the function of circulating IGF-1 and lead to a more comprehensive understanding of the biology of IGF-1 in health and vascular disease. 

## 7. Conclusions

Hepatic IGF-1 production decreases with age. This decline may contribute to brain aging and the progression of age-related neurological conditions, partially due to insufficient vascular remodelling and dysfunction. The reversible binding of IGF-1 to IGFBP-3 regulates the amount of bioavailable IGF-1 in circulation and maintains the homeostasis of IGF-1 function during ageing. As a metabolite from the binding fraction of IGF-1, cGP retains the same binding affinity to IGFBP-3 as IGF-1, thus collectively regulating the bioavailability of IGF-1 in circulation. An increase in cGP and/or a decrease in IGFBP-3 concentrations leads to more bioavailable IGF-1 in the plasma, whereas the decrease in cGP and/or the cGP/IGF-1 molar ratio and/or the increase in IGFBP-3 reduces the amount of bioavailable IGF-1, and thus IGF-1 function. This occurs, for example, in age-related cognitive impairment, poor stroke recovery, and hypertension. Therefore, the administration of cGP restores the equilibrium between IGF-1 and cGP, or simply displaces IGF-1 from the IGFBP-3 binding complex, thereby improving cognitive function, promoting recovery from stroke, and normalising blood pressure (in hypertensive rats). Further research in this area may well lead to the development of a plasma biomarker for monitoring not only the progression of vascular disorders, but also the effects of interventions in vascular conditions marked by IGF-1 deficiency. A similar regulatory process may play a minor role in brain, as IGF-1 in the brain is largely bioavailable (Figure 3).

## Figures and Tables

**Figure 1 molecules-28-01021-f001:**
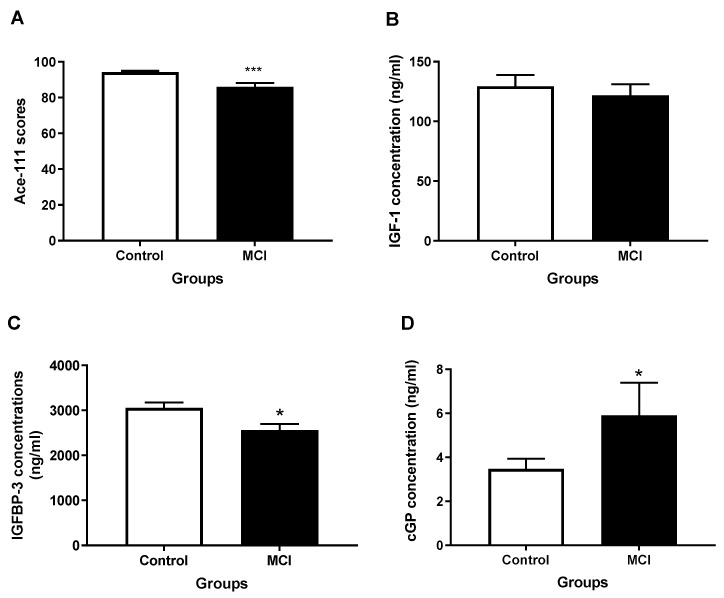
Changes in plasma concentrations of IGF-1 (**B**), IGFBP-3 (**C**), and cGP (**D**) in Alzheimer’s patients with mild cognitive impairment (**A**). The reduced IGFBP-3 and increased cGP concentrations represent a collective response to improve the bioavailability of IGF-1. * *p* < 0.05, *** *p* < 0.001.

**Figure 2 molecules-28-01021-f002:**
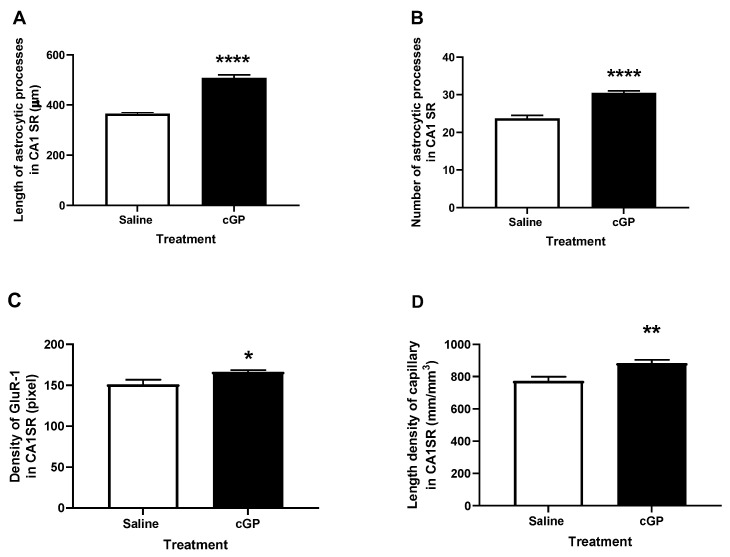
The maternal administration of cGP during brain development improves the astrocytic plasticity (**A**,**B**), glutamate trafficking (**C**), and vascular remodeling (**D**) of adult offspring. * *p* < 0.05, ** *p* < 0.01, **** *p* < 0.0001.

**Figure 3 molecules-28-01021-f003:**
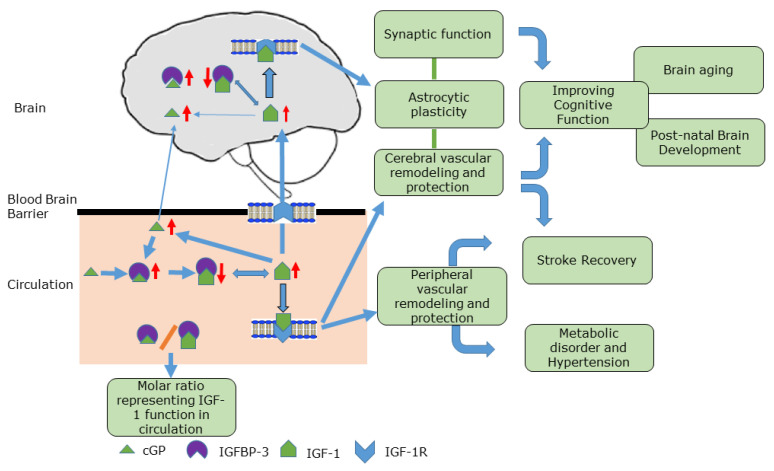
A graphic abstract showing the role of cGP in regulating the amount of bioavailable IGF-1 in circulation (pink area) and in brain tissues (grey area). These collectively improve IGF-1 operation in vascular function, and potentially provide beneficial effects on cerebral and systemic vascular diseases. The molar ratio of cGP/IGF-1 in plasma represents the amount of bioavailable IGF-1, and thus the function of IGF-1.

**Table 1 molecules-28-01021-t001:** Correlation between plasma concentration of IGFBP-3 and motor deficits.

Dependent Variable (Confounders)		cGP (ng/mL)	IGF-1 (ng/mL)	IGFBP-3 (ng/mL)
UPDRS (Age and MoCA)	B	−0.01	0.01	0.004
	P	0.98	0.60	**0.02**
	95%CI	−0.50, 0.48	−0.03, 0.05	0.001, 0.007
UPDRS (Age and Global Z)	B	0.20	0.004	0.004
	P	0.38	0.85	**0.02**
	95%CI	−0.25, 0.66	−0.04, 0.05	0.001, 0.007
UPDRS (Age and LMD)	B	0.26	0.01	0.004
	P	0.27	0.61	**0.01**
	95%CI	−0.21, 0.72	−0.03, 0.06	0.001, 0.01

UPDRS: Movement Disorder Society Unified Parkinson’s Disease Rating Scale; MoCA: Montreal Cognitive Assessment; LMD: learning memory domain scores; Global Z score: global measure of cognitive function derived from comprehensive neuropsychological testing.

**Table 2 molecules-28-01021-t002:** Differences in anxiety and depression scores before and after blackcurrant therapy.

		Before	After	*t*-Test	*p*
Total HADS	Mean Score	9.5	5.4	*t*_9_ = 3.45	**0.007**
	SD	6.8	3.9		
Anxiety Subscale	Mean Score	5.4	2.5	*t*_9_ = 2.69	**0.025**
	SD	3.6	1.8		
Depression Subscale	Mean Score	4.1	2.9	*t*_9_ = 2.03	0.074
	SD	3.5	2.7		

## Data Availability

Not applicable.

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
