# Peer review of "Cyclic Glycine-Proline (cGP) Normalises Insulin-Like Growth Factor-1 (IGF-1) Function: Clinical Significance in the Ageing Brain and in Age-Related Neurological Conditions"

_molecules, 2023, doi:10.3390/molecules28031021_

Round 1
Reviewer 1 Report
The association between cGP, a metabolite of a binding site of IGF-I, and IGF-I signaling is intriguing (Guan et al., Scientific Reports 2014). However there are major issues concerning this review in its present form.
1. The use of the term autocrine regulation is incorrect. Autocrine is a form of cell signaling in which a cell secretes a substance that binds to receptors on that same cell, leading to changes in the cell. This could pertain to the endothelial cell experiment in culture. Paracrine signaling might be more appropriate for discussion of brain IGF-I, and endocrine for circulating IGF. Circulating IGF-I and local brain IGF action can be very different. The different contexts need to be distinguished in this review.
2. The flow of the text is difficult to follow for several reasons: a) Because of the mixture of autocrine, paracrine and endocrine measures in the different citations; b) The references on IGF-I signaling are very early, e.g. 18/52 are from <2000 and 17/52 are from <2012 making 67% of the references from over 10 years ago. A lot has been learned and unlearned about IGF signaling since then.
3) Several citations are about the results of cGP administration alone having effects on systolic blood pressure, improving memory and aiding stroke recovery. Association with local IGF signaling seems tenuous.
4) cGP and IGF-I in the different clinical situations was difficult to follow. It would be helpful to the readers to have flow charts for the different conditions, especially PD. Endocrine and paracrine functions of cGP and IGF need to be distinguished. Is plasma IGF-I the main source of in brain tissue? IGFBP-3 is sometimes discussed as part of the endocrine function of cGP/IGF-I. IGFBP-2 is briefly cited as being associated with brain inflammation in PD. Was IGFBP-2 studies in any of the in vitro experiments.
Author Response
Comments of Reviewer 1:
The association between cGP, a metabolite of a binding site of IGF-I, and IGF-I signaling is intriguing (Guan et al., Scientific Reports 2014). However there are major issues concerning this review in its present form.
- The use of the term autocrine regulation is incorrect. Autocrine is a form of cell signaling in which a cell secretes a substance that binds to receptors on that same cell, leading to changes in the cell. This could pertain to the endothelial cell experiment in culture. Paracrine signaling might be more appropriate for discussion of brain IGF-I, and endocrine for circulating IGF. Circulating IGF-I and local brain IGF action can be very different. The different contexts need to be distinguished in this review.
Response to comment 1:
Presumably the comment of ‘autocrine regulation in cells’ refers to IGF-1 regulation in brain tissues. The process of ‘cell secretes a substance that binds to receptors on that same cell, leading to changes in the cell’ is a process that has been described as paracrine regulation of IGF-1. The review has briefly described this in lines 361-367. Autocrine regulation of IGF-1 has been best evaluated in circulation and plays a major role in regulating bioavailability of IGF-1 circulation with a minor role in brain tissues due to the majority of IGF-1 in brain is bioavailable. Given paracrine regulation of IGF-1 happens mainly under pathological conditions, IGF-1 regulation in brain tissues is often described as autocrine/paracrine processes.
Endocrine regulation of IGF-1 refers to the growth hormone regulated IGF-1 production, mainly in liver, which do not determine the bioavailability of IGF-1. This review focuses on a process that regulates the bioavailability of IGF-1 in circulation which has clearly recognised as an autocrine regulation.
- The flow of the text is difficult to follow for several reasons: a) Because of the mixture of autocrine, paracrine and endocrine measures in the different citations; b) The references on IGF-I signaling are very early, e.g. 18/52 are from <2000 and 17/52 are from <2012 making 67% of the references from over 10 years ago. A lot has been learned and unlearned about IGF signaling since then.

Reviewer 2 Report
Review of the manuscript entitled:
Cyclic glycine-proline (cGP) normalises insulin-like growth factor-1 (IGF-1) function: clinical significance in ageing brain and age-related neurological conditions
Here are some specific comments:
The strengths of the manuscript are that it is very well written making it very readable even for those not knowledgeable about this topic and its implications for neurological disease. The references are comprehensive. As the authors note, there is little in the published literature about the role of this molecule and the metabolic pathway in a particular disease and this review may foster more interest in pursuing research that may help to elucidate how it plays a role in pathophysiology of neurological disorders.
It is a review article as so there is no original research to critique and I cannot think of any significant weaknesses.
Whoever was the editor that initially reviewed this paper and approved it for submission for peer review should be commended. It is a well written review.
If, however the academic staff is not persuaded by my review, perhaps it should be sent to another reviewer
Author Response
We appreciate the comments from Reviewer 2 and hope the publication of this review will renew IGF-1 research in near future.
Reviewer 3 Report
The topic is interesting and clearly discussed. I think it can be a good contribution for the topic. I believe that to consider cGP as a plasma biomarker in age-related brain diseases several other cofactors need to be considered, especially for multifactorial neurodegenerative disorders. I hope it will be possible in the future.
I think authors should make the tables easier to read with a caption that clarifies the content.
Author Response
Our response:
We have added a graphic abstract which summarises the key points of the review
Reviewer 4 Report
In this review study, the authors evaluated the “Cyclic glycine-proline (cGP) normalises insulin-like growth factor-1 (IGF-1) function: clinical significance in ageing brain and age-related neurological conditions” The authors reported that cGP administration reduces systolic blood pressure, improves memory and aids stroke recovery. This study is relevant and interesting and should be accepted after major corrections.
- Improve section 1.1. Decline of IGF-1 function with age and its association with age-related neurological condition.
- Write a sentence to justify this caption “cGP is a bioactive compound”
- The authors should develop diagrammatic representative of possible mechanism linking the role of cGP and hypertension, stroke and cognitive impairment in neurological conditions.
- The clinical relevance needs to be revised; specific clinical relevance should be stated.
Author Response
Comments from Reviewer 4:
In this review study, the authors evaluated the “Cyclic glycine-proline (cGP) normalises insulin-like growth factor-1 (IGF-1) function: clinical significance in ageing brain and age-related neurological conditions” The authors reported that cGP administration reduces systolic blood pressure, improves memory and aids stroke recovery. This study is relevant and interesting and should be accepted after major corrections.
- Improve section 1.1. Decline of IGF-1 function with age and its association with age-related neurological condition.
Response to comment 1
The section has revised as:
1.1. Decline of IGF-1 function with age and its association with age-related conditions.
As an endogenous hormone, IGF-1 is essential in brain development, cognitive function, energy metabolism and wound healing [6]. In addition to its function as a neurotropic factor in brain, IGF-1 regulates cerebral and systemic vascular remodelling and function [7, 8]. Growth hormone (GH) controls liver production of IGF-1 [9], which declines with ageing to a very low measurable level in people over 60 years [9, 10]. An age-related deficiency in IGF-1 production and function can contribute to cardiac vascular aging [11], as well as cerebrovascular diseases such as stroke [5, 12, 13], hypertension [6,14] and cognitive impairment [15-17].
- Write a sentence to justify this caption “cGP is a bioactive compound”
Responses to comment 2
A sentence of ‘Following section addressed the discovery and research development of cGP as a endogenous neuropeptides.‘ has been added to the section.
To make it more logical 2 subtitles have added to this section as
2.1. Formation of endogenous cGP
2.2. cGP as a bioactive peptide
- The authors should develop diagrammatic representative of possible mechanism linking the role of cGP and hypertension, stroke and cognitive impairment in neurological conditions.
Response to comment 3
We have added a graphic abstract
- The clinical relevance needs to be revised; specific clinical relevance should be stated.
Responses:
The first sentence has been revised as ‘In general, the clinical relevance of cGP is in those conditions where there is IGF-1 deficiency, particularly the age related vascular diseases - metabolic hypertension, stroke, age-related cognitive decline - and some neurodegenerative disorders such as Parkinson’s disease (PD).’.
Round 2
Reviewer 1 Report
Disagree with the authors self-description of autocrine, paracrine, and endocrine regulation. Recommend replacement of these terms with local and circulating IGF-1 to increase clarity.
Please define terms in Table 1: UPDRS, MoCA, Global Z, LMD
Please clarify authorship for reference 84
Author Response
The title of section 1.2 has been revised as: ‘The regulation of bioavailability of IGF-1 in circulation’. The term of 'autocrine regulation' has been either removed or revised accordingly throughout. For example, 'the reversible binding of IGF-1 to IGFBP-3 in circulation' or 'reversible binding of IGF-1 to IGFBP-2 in brain tissues' or ‘the regulation of bioavailability of IGF-1’ or ‘regulating the amount bioavailable IGF-1’
The terms of UPDRS, MoCA, Global Z, LMD have been defined in Table 1
Biomarkers Definitions Working Group. has been added to the reference 84.